# Learning-to-Count by Learning-to-Rank: Weakly Supervised Object Counting & Localization Using Only Pairwise Image Rankings

## Abstract

Object counting and localization in dense scenes is a challenging class of image analysis problems that typically requires labour intensive annotations to learn to solve. We propose a form of weak supervision that only requires object-based pairwise image rankings. These annotations can be collected rapidly with a single click per image pair and supply a weak signal for object quantity. However, the problem of actually extracting object counts and locations from rankings is challenging. Thus, we introduce adversarial density map generation, a strategy for regularizing the features of a ranking network such that the features correspond to an object proposal map where each proposal must be a Gaussian blob that integrates to 1. This places a soft integer and soft localization constraint on the representation, which encourages the network to satisfy the provided ranking constraints by detecting objects. We then demonstrate the effectiveness of our method for exploiting pairwise image rankings as a weakly supervised signal for object counting and localization on several datasets, and show results with a performance that approaches that of fully supervised methods on many counting benchmark datasets while relying on data that can be collected with a fraction of the annotation burden.

## 1 Introduction

Object counting is a popular computer vision problem that involves localizing and quantifying the number of objects within an image, which has broad applicability across several domains. For example, counting problems have been explored throughout plant analysis (David et al., 2020; 2021; Häni et al., 2020; Bargoti & Underwood, 2017; Minervini et al., 2015; Teimouri et al., 2018), wildlife population monitoring (Arteta et al., 2016), crowd surveillance (Change Loy et al., 2013; Chen et al., 2013; Loy et al., 2013; Chen et al., 2012; Zhang et al., 2016; 2015), tissue sample analysis (Marsden et al., 2018; Paul Cohen et al., 2017; Xie et al., 2018; Kainz et al., 2015), and vehicle surveillance (Guerrero-Gómez-Olmedo et al., 2015; De Almeida et al., 2015; Wang et al., 2008). However, we highlight two major problems that prevent counting problems from being widely applicable. First, there is no consensus on how to structure the optimization target. Different methods have approached the problem using a variety of annotations, which include bounding boxes (Ren et al., 2015; Redmon et al., 2016), global object counts (Chattopadhyay et al., 2017), point maps (Laradji et al., 2018), and density maps (Lempitsky & Zisserman, 2010). Bounding boxes are a popular optimization target for detection problems. However, these annotations tend to under-perform on counting problems, in particular, when scenes contain many highly occluded objects at a variety of scales (Chattopadhyay et al., 2017). Density map annotations often perform better, but suffer from a lack of information related to object characteristics such as scale (Arteta et al., 2016). Second, as the quantity and density of objects in an image increase, all of the above annotation types become labour intensive to collect and prone to significant annotator noise (Arteta et al., 2016). We circumvent the burden of object labelling by introducing a simple form of annotation that can be rapidly collected, and design a method to propose density maps from these annotations.

We propose using pairwise image ranking, a binary valued annotation that orders image pairs based on per-image object counts. Previous work has demonstrated these annotations to be an effective training signal for semi-supervised counting problems (Liu et al., 2018). Whereas this work relied

on leveraging automatically collected intra-image relationships between the whole image and sub-image crops, we extend this idea further by proposing a method for learning to count and localize objects exclusively from inter-image pairwise image rankings. Our inter-image annotations carry a weak but information rich training signal despite being quick to collect. Humans are adept at rapidly assessing which of two images has more objects of interest, and untrained annotators are capable of discriminating between two images at a ratio of 10:11 objects in under 0.75 seconds (Pica et al., 2004; Halberda & Feigenson, 2008). While previous optimization targets require an annotator to identify and count out each object, pairwise image rankings only require a single click from an annotator. In Tables 2, 3, and 4 we evaluate the annotation burden for recent object counting strategies and demonstrate that our annotation strategy carries only a fraction of the burden. Given these savings, there is significant value in developing methods that learn to extract counts and locations from this type of training data.

The major technical challenge in exploiting this type of annotation is finding a way to extract counts and locations given only pairwise rankings. In this work, we propose an adversarial strategy for regularizing the penultimate representation of a ranking network to have the properties of a density map by comparing it to a pseudo-density map distribution. This places a soft integer and localization constraint on the representation. We argue that this strategy forces the model to solve the ranking sub-task by detecting re-occurring objects that can satisfy all of the ranking constraints. Combined with the weak quantity signal provided by pairwise image ranking annotations, this strategy produces a counting and localization model that is competitive with architectures trained on images with their density maps and global counts annotations. In summary, our work makes the following contributions:

1. We propose object-based pairwise inter-image ranking as a novel, low-cost annotation strategy for weakly supervised counting and localization, and demonstrate that it performs comparably with fully supervised counting methods.

2. We propose adversarial density map regularization, a method for enforcing that the network output has the properties of a density map.

## 2 Previous Work

**Object Counting with Limited Data.** Object counting methods perform best when learning from density maps (Lempitsky & Zisserman, 2010; Chattopadhyay et al., 2017), which carry a high annotation burden. Several methods have emerged that attempt to eliminate this burden. These methods can be loosely split into 4 categories – semi-supervised, knowledge transfer, sample selection, and weakly supervised methods.

Semi-supervised counting methods alleviate the annotation burden by including additional unlabelled data; a recent approach (Liu et al., 2019b; 2018) introduced an unsupervised ranking loss that exploits the fact that any image has as many or more objects than any cropped sub-portion of that image. Other methods (Sam et al., 2019) have proposed more general feature learning strategies. Knowledge transfer methods explore transferring features between counting problems; one method (Zhang et al., 2015) proposed learning from a multi-modal dataset which contained a mix of density maps and global object counts. A more recent approach (Ranjan et al., 2021) proposed a few-shot learning strategy where the model was trained using exemplar and density map pairs such that it could be extended to novel object classes. Active learning methods (Ranjan et al., 2020; Zhao et al., 2020) approach the problem by finding ways to only label the most important examples in a new domain. Weakly supervised methods have explored strategies for better utilizing global counts as annotations. Recent methods (Yang et al., 2020; Lei et al., 2021) proposed various regularization terms. One such method introduced a soft-sorting multi-task loss (Yang et al., 2020), which involved learning from global object count annotations directly and indirectly through a soft-sorting task. Another recent work (Liang et al., 2021) has suggested that self-attention can improve learning from global counts. A different approach (Cholakkal et al., 2019) involved using object recognition combined with global object counts to create class-specific density maps. Our method differs from the above weakly-supervised methods by learning exclusively from a much weaker signal than global object counts.

**Weakly-Supervised Object Localization.** Related to weakly-supervised object counting is weakly-supervised object localization (WSOL) (Zhang et al., 2021). This task involves localizing objects using only some weak image-level label. Arguably, the most important WSOL methods are class activation maps (CAM) (Zhou et al., 2016) and Grad-CAM (Selvaraju et al., 2017). The authors of CAM discovered that using global average pooling in a classification network allowed them to localize parts of the object being classified. Grad-CAM extends this work by re-weighting the activation map using the gradients of some target class. Separately, image region erasing has become a popular strategy for learning localized features from global class labels. Some works (Singh & Lee, 2017; Choe & Shim, 2019) randomly erase image regions, forcing the network to rely on the whole object to classify the image. Other works (Choe & Shim, 2019; Zhang et al., 2018; Kim et al., 2017; Wei et al., 2017) proposed adaptive erasing strategies for removing the image regions that contribute most to classification, forcing the network to explore other regions in an attempt to accurately classify the image. Our problem shares similarities with WSOL in that we are using a global image label, albeit a pairwise image ranking label and not an image class label, to detect objects.

**Counting, Detection, and Output Representations.** Localizing and counting target objects in images is an important computer vision problem. However, there is no consensus on the appropriate optimization target for this task. Bounding boxes (Ren et al., 2015; Redmon et al., 2016) are a popular approach, but regressing over box coordinates is a difficult problem that underperforms on counting tasks (Chattopadhyay et al., 2017). Density maps (Lempitsky & Zisserman, 2010) are frequently used as an alternative, but, in their basic form, suffer from a lack of information about object scale and scene geometry. To remedy this, different strategies for producing adaptive density maps have been explored. Self-attention (Wan & Chan, 2019) for selecting the best density map scale has worked for crowd counting problems. Similarly, ad-hoc geometric priors (Zhang et al., 2016) have been used to fuse scene geometry and density maps. Other methods have explored finding object blobs from dot maps (Laradji et al., 2018); multi-scale and multi-resolution density maps (Idrees et al., 2018); and learning directly from dot maps by treating density map estimation as a perspective-guided unbalanced optimal transport problem (Wan et al., 2021).

**Ranking as Supervision** The use of ranking as a training signal, sometimes known as ordinal regression, originates within information retrieval research. RankNet (Burges et al., 2005), a document retrieval network, emerged as the first deep learning approach to ranking. However, this approach has been extended into several computer vision applications. Facial age estimation (Chen et al., 2018; Lim et al., 2020) has benefited from pairwise image rankings, which learn the 'amount' of age in an image. Pairwise image ranking has been used to localize facial attributes (Singh & Lee, 2016), compare agent skills in videos (Doughty et al., 2018), and detect video highlights (Yao et al., 2016). Ranking has also been used for ranking object proposals and class label proposals (Li et al., 2017; Liu et al., 2021). These applications highlight that ranking as an optimization signal plays a significant role in computer vision applications.

## 3 PAIRWISE RANKING AS WEAK SUPERVISION

Pairwise image ranking is a task which requires an annotator to provide an ordering for image pairs based on the quantity of objects present in the image. Given some dataset

$$\mathcal{D}_{rank} = \{(x_i, x_j), r_{ij} \equiv c_i \geq c_j\}^N,$$

we would like to extract the underlying object counts $c_i$ and $c_j$ given only the image ranking constraints $r_{ij}$ for $N$ pairs of images $(x_i, x_j)$. The goal of this paper is to develop a strategy to use the implicit weak object quantity signal present in $\mathcal{D}_{rank}$ to solve the counting task. However, we first provide a motivation for this strategy by exploring the limitations of other annotation strategies and the strengths of pairwise image ranking.

Density maps are the most popular annotation strategy for solving counting problems. Here, a density map is defined as a heatmap where a Gaussian kernel is convolved with point-wise annotations, with the property that the density map integrates to the global count. This annotation formulation is useful because it provides a location-based target for the object counting problem but has less complexity than bounding-box annotations. However, collecting point-wise annotations presents

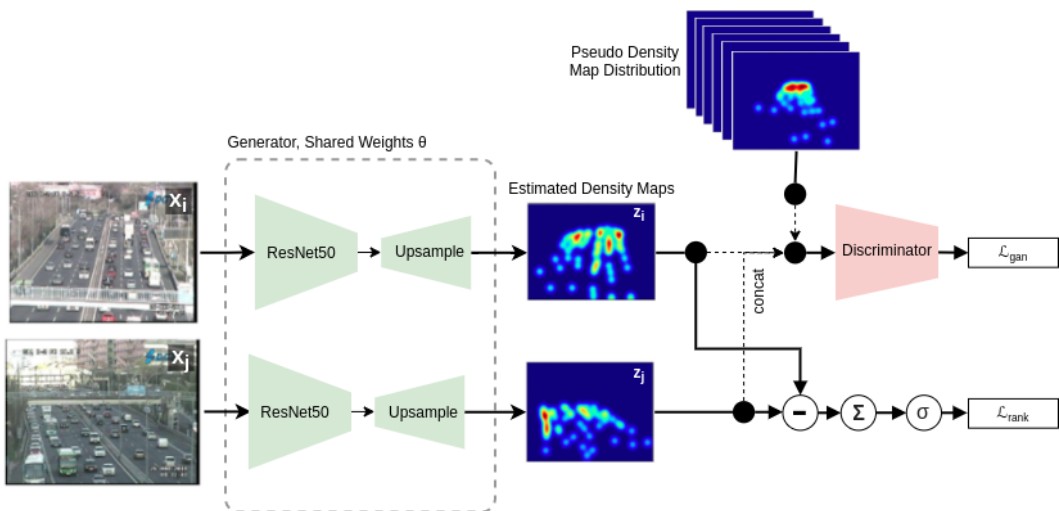

Figure 1: Method overview. Pairs of images with object count ranking labels are used to train a neural network to count and localize objects. Each image $x_i$ and $x_j$ are passed to the generator model and their respective outputs $z_i$ and $z_j$ are used to calculate $\mathcal{L}_{rank}$, which provides the weak object quantity training signal. In addition to this, we include an adversarial density map generation loss, $\mathcal{L}_{gan}$, which encourages the output of the generator model to have the properties of a density map.

significant challenges. Strategies that learn directly from global object counts are able to reduce the annotation burden associated with density maps, but these methods still carry their own significant annotation burden. Global counting annotations and point-wise annotations require an annotator to visually localize all objects and record those labels. This can be time consuming and error prone, especially as object quantity, occlusion, and density increase. In fact, it has been demonstrated (Arteta et al., 2016) that when multiple annotators label an image with point-wise annotations, the annotators often suffer from fatigue or make inconsistent decisions when faced with ambiguity and make regular errors that lead to significant signal noise. Given the annotation burden that exists with all of the above methods, it would be valuable to establish a weak signal that could be rapidly collected by annotators while still leading to similar performance on the counting problem.

Researchers in human psychology have found that humans can rapidly assess which of two groups of objects has the most objects if the difference in object count is greater than a particular ratio. This ratio is approximated by Weber-Fechner law, which is a law that describes the change in a stimulus necessary for some human to perceive the difference relative to the existing stimulus. For the task of pairwise image ranking, researchers (Pica et al., 2004; Halberda & Feigenson, 2008) have found that within 0.75 seconds, untrained adults are capable of determining which of two images has the most objects if the object count ratio is smaller than approximately 9:10 and 10:11 (independent of the absolute count). This suggests that, for untrained adults, the task of rapidly annotating image pairs with their respective rank should be relatively simple as long as the ratio between object counts within the image pairs remains smaller than the given ratio. Thus, given a weak signal that can be rapidly collected, we are left to answer the difficult question of how to extract the counts.

## 4 METHOD

The goal of our method is to develop a model which can extract object counts given only pairwise image rankings. These annotations contain a weak signal for object quantity, which we use to train a neural network, $z_i = f(x_i; \theta)$, outlined in Figure 1. However, this target alone is not enough to learn a representation from which we can extract global object counts. We solve this problem by proposing adversarial density map estimation, a strategy which structures the output of $f_\theta$ to have the properties of a density map. With this strategy, our model learns to count objects and even localize those objects within an image.

## 4.1 RANKING NETWORK

The purpose of the network $f(x_i; \theta)$ is to extract the underlying weak object quantity signal from the pairwise image ranking annotations. As outlined in Figure 1, our base counting model $f_\theta$ receives two images as input, $x_i$ and $x_j$, and outputs representations $z_i, z_j \in [0, 1]^{56 \times 56}$. We use these representations to model the probability that $c_i \geq c_j$ by approximating the true distribution as follows:

$$p_{rank} = P(r_{ij}|x_i, x_j; \theta) = \sigma(\sum_{k,l} (z_i)_{kl} - (z_j)_{kl}), \tag{1}$$

where $\sigma$ is the sigmoid operation and $(k, l)$ are the indices for the representations $z_i$ and $z_j$. Here, we benefit from the fact that when the difference between $z_i$ and $z_j$ is positive, the sigmoid operation outputs a value greater than 0.50. Whereas when the difference is negative, the sigmoid operation outputs a value less than 0.50. This allows us to model which of two images in a pair has more objects by inspecting the magnitudes of the sum over $z_i$ and $z_j$. Thus, by optimizing $\theta$ using the following loss function:

$$\mathcal{L}_{\text{rank}} = -\mathbb{E}_{p_{data}}[\log(p_{rank})], \tag{2}$$

the model must learn to minimize the number of pairwise inversions (from the ground truth distribution $p_{data}$) among all ranking examples in the training dataset, which creates a partial ordering of all the images by object count. However, this ordering is decoupled from any notion of object identity and location. In the next section, we propose an approach to explicitly connect the output representation to object locations.

## 4.2 ADVERSARIAL DENSITY MAP GENERATION

Previous empirical results have demonstrated the value of density maps as a location-based annotation for counting problems. Density maps are structured such that they place Gaussian density where objects occur and integrate to the global count. These properties are useful because they explicitly connect the detection and counting task. While we do not have access to density maps, we argue that allowing the counting network $f_\theta$ to propose density maps when solving the pairwise image ranking problem captures some of the useful properties of density maps. We explore a strategy for structuring the output representation $z_i$ to have these properties.

We start by establishing a pseudo point map distribution from which we can randomly sample point maps $z_{\text{point}} \in \{0, 1\}^{56 \times 56}$. We convolve $z_{\text{point}}$ with $K_\delta$, a 2D kernel with width $\delta$, to generate a pseudo density map:

$$\tilde{z}_{\text{dmap}} = K_\delta * z_{\text{point}}. \tag{3}$$

If an appropriate sampling distribution is chosen, then we can establish an adversarial training objective that penalizes the network output $z_i$ when it deviates from the properties of a density map. We first describe the adversarial training objective, and then describe methods of selecting appropriate pseudo density map distributions.

Adversarial training (Goodfellow et al., 2014) is a widely adopted technique for modeling the underlying generating distribution that explains a dataset. This training strategy involves optimizing two neural networks, our counting network $f$ (termed the generator) and a discriminator $D$. The generator is tasked with generating samples from an underlying distribution and the discriminator is tasked with evaluating whether a sample came from the pseudo point map distribution or the generator's distribution. The generator is optimized using feedback from the discriminator, and the discriminator is optimized using the samples from the generator. Under this training paradigm, the discriminator $D : \mathbb{R}^{56 \times 56} \to \mathbb{R}$ predicts whether its input is a sample from the pseudo-density map distribution. We use the LS-GAN objective function (Mao et al., 2017), which is given as:

$$\mathcal{L}_{gan}^{f} = -\mathbb{E}_x \left[ D(f_\theta(x_i) - 1)^2 \right], \tag{4}$$

for the generator, and:

$$\begin{aligned} \mathcal{L}_{gan}^{D} = -& \mathbb{E}_{\tilde{z}_{\text{dmap}}} \left[ (D(\tilde{z}_{\text{dmap}}) - 1)^2 \right] \\ & + \mathbb{E}_x \left[ D(f_\theta(x_i))^2 \right]. \end{aligned} \tag{5}$$

Selecting a point map distribution is a challenging problem, as it is known that convolution neural networks can encode spatial information (Islam et al., 2020). Given this, a discriminator can penalize the underlying spatial distribution of objects learned by the generator. Given this situation, we choose to sample point maps uniformly. To produce the pseudo point map distribution, first, we uniformly sample a total count, $c_{pseudo}$, for the number of Gaussian blobs in a particular density map:

$$c_{pseudo} \sim \mathcal{U}_{\{0, N_c\}}, \tag{6}$$

where $N_c$ is the estimated maximum object count for the dataset. Then, we uniformly sample $c_{pseudo}$ co-ordinates:

$$i, j \sim \mathcal{U}_{[0,56] \times [0,56]}, \tag{7}$$

which gives us $z_{\texttt{point}}$. This uninformative prior makes it more difficult for the discriminator to exploit the underlying spatial distribution, as any point location is equally likely.

### 4.3 Implementation Details

**Architecture.** We use ResNet50 (He et al., 2016) as the default architecture for the underlying counting model $f_\theta$. The ResNet50 model shares parameters across both branches of the network in Figure 1. We generate a density map $z_i$ in a similar fashion to previous work on segmentation that use fully convolutional networks (Long et al., 2015). First, we pass an image $x_i$ through ResNet50 and collect features at different resolutions throughout the network. Then, we up-sample the features using transposed convolutions and combine these features to form $z_i$, a $56 \times 56$ density map estimate. To construct the ranking network, we pass two images through $f_\theta$ and then we take the sum of the difference between the respective density map estimates before passing the final number through a sigmoid activation whose output is used as the predicted image ranking.

**Adversarial Setup.** The density map estimates, $z_i$ and $z_j$, are passed to the discriminator. The discriminator predicts whether or not the resulting density map estimates look like samples of the pseudo density map distribution. The discriminator itself is designed to have a relatively simple architecture, comprised of 5 convolutional layers with $4 \times 4$ filters and LeakyReLU activations that downsample the density map. Then, these features are passed through a final fully connected layer.

**Training.** Each model for each experiment is trained for 200 epochs using the Adam optimizer with a batch size of 32. The learning rate is randomly selected between $10^{-5}$ and $10^{-3}$, with $5 \times 10^{-5}$ being a reliable choice. Similarly, the $\beta_1$ parameter of the Adam optimizer was randomly selected from a normal distribution, with $0.50$ being the most reliable choice. The optimizer hyperparameters are shared by both the ranking architecture and the discriminator. Each image is augmented using a random combination of horizontal flipping, rotation, colour jitter, and noise such that no objects are lost and such that the object identity is preserved. Due to the instability of GANs, we run multiple experiments and select the best performing model using the validation set.

## 5 Experiments

### 5.1 Datasets

We benchmark our results on four object counting datasets, which we describe here. TRANCOS (Guerrero-Gómez-Olmedo et al., 2015) is a 2015 vehicle counting benchmark dataset containing 1,244 images with point map annotations for 46,796 highly occluded vehicles in traffic. Penguins (Arteta et al., 2016) is a 2016 animal counting benchmark dataset containing around 82,000 images of penguin colonies taken under a diverse set of environmental conditions across 40 different locations. Each image contains several dot maps from different annotators. The UCSD dataset (Chan & Vasconcelos, 2008; Chan et al., 2008) is a 2008 crowd counting dataset containing 2,000 video frames of pedestrian traffic across a campus. The Mall dataset (Change Loy et al., 2013; Chen et al., 2013; Loy et al., 2013; Chen et al., 2012) is a 2013 crowd counting data containing 2,000 video frames of pedestrian traffic in shopping malls, with over 60,000 human heads annotated across the dataset. All of these datasets are challenging benchmarks as they contain highly occluded objects in dense scenes with a variety of environmental conditions and scales.

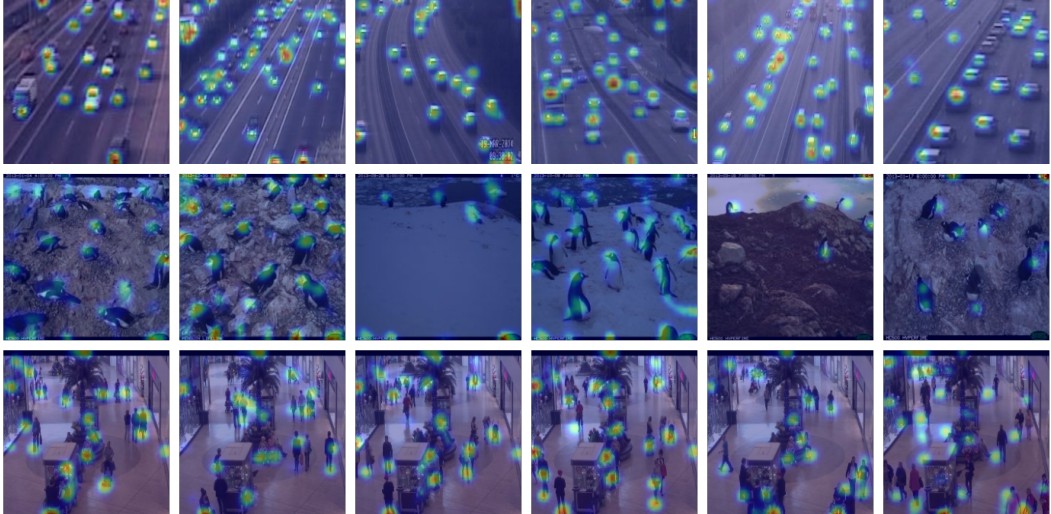

Figure 2: Qualitative examples of density maps predicted by the baseline model at inference time. Top: Trancos. Middle: Penguins. Bottom: MALL.

Table 1: Ablation of results for the baseline model modified by removing different components to reveal their underlying contribution to the model

| Method | Trancos | | Mall | | Penguins | | UCSD | |
|---|---|---|---|---|---|---|---|---|
| | MAE | $R^2$ | MAE | $R^2$ | MAE | $R^2$ | MAE | $R^2$ |
| $\mathcal{L}_{\texttt{rank}} + \mathcal{L}_{\texttt{gan}}$ | **5.89** | **0.76** | **3.62** | **0.41** | **7.43** | 0.52 | **5.04** | **0.47** |
| $\mathcal{L}_{\texttt{gan}}$ | 13.19 | -0.15 | 4.80 | 0.00 | 13.70 | -0.08 | 7.70 | -0.45 |
| $\mathcal{L}_{\texttt{rank}}$ | 9.47 | 0.51 | 5.46 | -0.14 | 7.72 | **0.58** | 7.04 | -0.15 |

### 5.1.1 SAMPLING IMAGE RANKING DATA

There are presently no well-established image ranking datasets available for weakly supervised object counting and localization benchmarking. Given this, all image ranking datasets used for evaluating our experiments must be curated. We experiment with the object counting and detection datasets outlined above, and reformulate all of the available datasets as ranking datasets as follows. Given a counting dataset

$$\mathcal{D}_{count} = \{x_i \in \mathbb{R}^{h,w,d}, c_i \in \mathbb{N}\}_{i=0}^{N_c},$$

where $d$ is the number of channels and $c_i$ is the object count in image $x_i$, we randomly sample $N = 2,000$ image pairs $(x_i, x_j)$ and calculate their pairwise ranking as $r_{ij} = c_i \geq c_j$. This provides us with a curated ranking dataset:

$$\mathcal{D}_{rank} = \{(x_i, x_j)_n, (r_{ij})_n \equiv (c_i \geq c_j)_n\}_{n=1}^{N}.$$

For our experiments, we impose no constraints on the sampling procedure and generate the training ranking dataset by simply uniformly sampling examples from the training dataset. We benchmark against the original counting test data provided with each dataset.

### 5.2 RESULTS

**Ablation of Model Components.** In Table 1, we explore the contribution of each model component to the test error. We evaluate each experiment using MAE and $R^2$. MAE is the mean absolute error and measures the difference between the predicted count and ground truth count, with a lower

score corresponds to a better performing model. $R^2$ is the coefficient of determination, which describes how well the model fits the data with a higher score corresponding to a better fit. When only $\mathcal{L}_{gan}$ is included during model training, we find that the model fails to detect and count objects. This result is intuitive, as the weak quantity signal is only provided by $\mathcal{L}_{rank}$. However, when only $\mathcal{L}_{rank}$ is included during model training, we find that the model under-performs when compared to the model trained with both $\mathcal{L}_{rank}$ and $\mathcal{L}_{gan}$. This result demonstrates that $\mathcal{L}_{gan}$ contributes an important training signal when solving the object counting problem using pairwise image rankings.

**Evaluating the Annotation Burden vs. Error Trade-Off.** We compare our results with state-of-the-art object counting methods and we provide an estimate of the annotation burden for each

Table 2: Comparison of the test error and annotation time for state-of-the-art counting methods on the TRANCOS crowd counting dataset. When methods use the standard dot-map training set, their annotation times are equivalent.

| Method | Supervision | Est. Annotation Time | MAE |
|---|---|---|---|
| Hydra CCNN (Oñoro-Rubio & López-Sastre, 2016) | Dot-map | 9 hr 11 min | 10.99 |
| FCN-MT (Zhang et al., 2017b) | Dot-map | 9 hr 11 min | 5.31 |
| FCN-HA (Zhang et al., 2017a) | Dot-map | 9 hr 11 min | 4.20 |
| LC-PSPNet (Laradji et al., 2018) | Dot-map | 9 hr 11 min | 3.57 |
| CSRNet (Li et al., 2018) | Dot-map | 9 hr 11 min | 3.56 |
| SPN (Chen et al., 2019) | Dot-map | 9 hr 11 min | 3.35 |
| ADSCNet (Bai et al., 2020) | Dot-map | 9 hr 11 min | 2.60 |
| Glance (Chattopadhyay et al., 2017) | Counts | 2 hr 41 min - 7 hr 54 min | 7.00 |
| Adv. Dmap (Ours) | Pairwise Rank | 25 min | 5.89 |

Table 3: Comparison of the test error and annotation time for state-of-the-art counting methods on the MALL crowd counting dataset.

| Method | Supervision | Est. Annotation Time | MAE |
|---|---|---|---|
| CNN-Boosting (Walach & Wolf, 2016) | Dot-map | 7 hr 13 min | 2.01 |
| LC-PSPNet (Laradji et al., 2018) | Dot-map | 7 hr 13 min | 2.01 |
| AL-AC (Zhao et al., 2020) | 10% w/ Dot-map | 43 min | 3.80 |
| Adv. Dmap (Ours) | Pairwise Rank | 25 min | 3.62 |

Table 4: Comparison of the test error and annotation time for state-of-the-art counting methods on the USCD crowd counting dataset.

| Method | Supervision | Est. Annotation Time | MAE |
|---|---|---|---|
| ADCrowdNet (Liu et al., 2019a) | Dot-map | 6 hr 59 min | 0.98 |
| Sorting (Yang et al., 2020) | Counts | 2 hr 00 min - 5 hr 54 min | 1.80 |
| Adv. Dmap (Ours) | Pairwise Rank | 25 min | 5.04 |

method. Here, we establish an annotation burden estimate for dot-maps and global object counts calculated over the training and validation set and compare it to the estimated annotation burden of our method. To estimate the annotation burden for dot-maps, we use the per-object annotation time of 1.1 s established by Cholakkal et al. (2020) and multiply this by the number of objects in the dataset. To estimate the annotation time for global object counts, we use a slightly more complex formula which includes the human ability to rapidly count objects within the range of 1 to 4, often referred to as the subitizing range. Saltzman & Garner (1948) established a counting speed of 0.1 s for each object within the subitizing range and 0.35 s for each additional object outside of the range. However, the participants in this experiment were only asked to count simple shapes placed on a white background. Cholakkal et al. (2020) looked more closely at human counting of complex object classes in realistic scenes and established a counting speed of 0.5 s within the subitizing range and 1.0 s for objects outside of this range. We use these two measures to create a range for our estimate of object counting speed. We calculate the annotation speed by analyzing the per-image count in each dataset and assigning a lower and upper bounds for the counting time of $[0.1, 0.5]$ seconds for each object within the subitizing range and $[0.35, 1.0]$ seconds for objects outside of the range. This provides our lower bound and upper bound for global object counting. To calculate pairwise image ranking speed, we use the per image-pair ranking time of 0.75 established by Pica et al. (2004); Halberda & Feigenson (2008) and multiply it by 2,000, which is the number of image-pairs in our sampled ranking dataset.

Table 2 compares our method to previous state-of-the-art counting methods evaluated on the TRAN-COS dataset, where we find that our method performs similarly to the method proposed by Zhang et al. (2017b), despite their method being supervised by dot maps requiring $\times 22$ the annotation time. More recent methods, such as the method proposed by Bai et al. (2020), outperform ours by a mean error of 3.29 vehicles per images, where each image contains an average of 38 vehicles. However, we find that our method requires 4.54% of the annotation time as the best performing fully supervised methods. We also find that our method outperforms *Glance* (Chattopadhyay et al., 2017), which learns from image-level object counts, while also requiring a smaller annotation burden (by a factor of 15.5% to 5.27%). Likewise, Table 3 presents the same comparison evaluated on the MALL dataset. We find that our method performs comparably to current state-of-the-art counting methods (our method's MAE differs by -0.18 to 1.61 object count), while requiring a fraction (5.77% to 5.81%) of the annotation time. Further, we find that our method outperforms the method proposed Zhao et al. (2020), which was specifically developed to deal with the annotation burden, while still only requiring a fraction (5.81%) of the annotation time. In Table 4, we evaluate our method on the USCD crowd dataset. We find that our method performs modestly when compared to state-of-the-art method (our MAE differs by 3.24 to 4.06 objects), but still requires a significantly smaller annotation burden (by a factor of 5.97% to 20.8%). We argue that these results demonstrate the value of pairwise image-ranking as a weak object counting signal and the value of our method for extracting object counts while minimizing the annotation burden.

## 6    CONCLUSION

In this paper, we present a weakly supervised strategy for extracting object counts and locations from pairwise image rankings, which can be easily and rapidly collected by annotators. We demonstrate that our method performs well on various benchmarks and approaches fully supervised baselines. This work highlights the value in exploring novel weak annotation formulations and provides a direction forward for solving counting problems in domains where annotation collection would otherwise be impermissible.

## 7    ETHICS STATEMENT

Object counting methods are broadly useful computer vision methods that are considered highly valued across several industries (see Section 1). However, object counting is also frequently used in human surveillance and thus lowering the burden necessary to develop new surveillance applications poses serious ethical concerns. In particular, tracking human movement can directly aid in military and carceral system applications, which can potentially empower unethical actors in the violation of fundamental human rights. While this should not inherently dissuade researchers from developing

new techniques for this application, it does call on the community to remain vigilant about which end-users benefit from the technology.

## 8    REPRODUCIBILITY STATEMENT

In section 4.3, we make an effort to detail network architecture, hyperparameters, and training parameters necessary to achieve the provided results. In sections 5.1 and 5.1.1, we detail the benchmark datasets used within our experiment and the strategy used to generate the ranking dataset. Additionally, we anonymously provide the source code necessary for reproducing the results.

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
