# OpenReview forum: "Learning-to-Count by Learning-to-Rank: Weakly Supervised Object Counting & Localization Using Only Pairwise Image Rankings"
_ICLR.cc/2022/Conference — ICLR 2022 Submitted_

### Official Review · Reviewer_yR54 · 2021-10-25

**Correctness:** 2
**Technical Novelty And Significance:** 2
**Empirical Novelty And Significance:** 3
**Recommendation:** 3
**Confidence:** 5

**Details Of Ethics Concerns:**

I do not have any concern about ethics.

**Main Review:**

Strength:
1. The weakly supervised setting could reduce the annotation cost and still achieve acceptable performance in several object counting benchmarks under specific conditions .

2. The proposed adversarial density map regularization technique could further improve the performance.

Weakness:
1. Novelty: The main contribution of this work is to use ranking information in object counting, as explained by the authors, ranking information is already used in previous object counting work[1,2], this work only extends  it to inter-image setting, which requires extra annotation while the original intra-image ranking is purely self-supervised[1,2] without extra annotation information ( in [1,2] the supervised loss is also used but the ranking loss itself is purely self-supervised ). Therefore, the novelty of this paper seems to be trivial.

2. Supervision: The pairwise ranking weak supervision can be highly unreliable in real world applications. For example for two images with both people and trees, and the first image contains both more people and trees than the other one, therefore, the ranking information can be vague as the model may not know which object to be counted, people or trees?

3. Adversarial Density Map: the point maps are sampled uniformly thus does not really capture the location information, I am a bit surprised that such pseudo point map could really help as it does not bring extra information to the network.

4. Annotation Cost: This pairwise ranking supervision has less annotation cost if the number of pair used is similar as the number of images in the dataset. However, if we have 500 images in the training set and there are can be up to 124,750 different pairs, then the most efficient way to obtain the ranking information is to annotate object number of all the images. Besides, if the object number is similar, it is hard to get the ranking information. Therefore, I think this setting does not generalize well in a broader setting.

5. Experiments: Missing evaluation with dense crowd counting benchmarks, for example ShanghaiTech PartA, UCF_QNRF, NWPU and UCF_CC_50 that are often followed by previous work and could be a good example to show the generalization of the proposed approach. Without such experiments, we cannot judge the generalization of the proposed approach.





[1]Xialei Liu, Joost Van De Weijer, and Andrew D Bagdanov. Leveraging unlabeled data for crowd counting by learning to rank. In Proceedings of the IEEE Conference on Computer Vision and Pattern Recognition, pp. 7661–7669, 2018.

[2]Xialei Liu, Joost Van De Weijer, and Andrew D Bagdanov. Exploiting unlabeled data in cnns by self-supervised learning to rank. IEEE transactions on pattern analysis and machine intelligence, 41(8):1862–1878, 2019.









------------------------------Post Rebuttal Comments--------------------------------------------------------------------------------------



I would like to thank the authors for providing the rebuttals, however, as explained in the post rebuttal comments, it does not solve my concerns and actually shows the limitation of this work. I therefore choose to keep my original ratings.

**Summary Of The Paper:**

In this work, authors propose to learn to count objects in weakly supervised setting where only pairwise ranking information is used as supervision. The annotation cost is expected to be much lower than the widely-used dot annotation. Besides, an adversarial density map regularization method is proposed to enforce the output remain the properties of a density map.

**Summary Of The Review:**

This paper propose an interesting setting of pairwise ranking information as weak supervision to count objects. However, as I mentioned in the main review part, there are several limitations in both methodology and experiments. I hope the authors could address my concerns listed above and I think we could not accept this paper at current stage.

---

> ### Author Response · Authors · 2021-11-20
> **Rebuttal**
>
> Hello reviewer! Thank you so much for your detailed and insightful review! We would like to respond to the points you have raised.
>
> Novelty: With regards to the works of Liu, et al., which you referenced in [1,2], their work leverages the fact that, given an unlabelled image and a cropped portion of that image, the crop must have less-than or equal the number of objects when compared to the whole image (intra-image ranking pairs). However, this self-supervision is not the only source of supervision that they use; their method also relies on fully supervised density maps which capture target object locations. The authors readily admit that their method does not work without the training signal provided by the density maps and provide an experiment where they perform pre-training with the intra-image rankings and fine-tune the model using density maps. They then perform the same fine-tuning experiment with an ImageNet pre-trained model and find that the intra-image ranking model underperforms by comparison. This suggests that intra-image ranking is a poor supervisory signal in isolation. The authors identify that this setup leads to trivial solutions, as there is no specified target object that determines the ranking. For example, there may be a few really simple patterns in the full image that are also captured in the crop. Then, the model can simply re-identify those patterns. The novelty of our work is that we are able to extract object counts and locations using only inter-image rankings and without any density maps.
>
> Supervision: Consider your example with trees + people. There is only ambiguity if you have a single image pair. However, if you have N examples containing ranking constraints and image pairs, the only way for that ambiguity to persist is if the numbers of both trees and people occur in the exact same inequality (rank) across every single example where they occur. Imagine that you have access to some model that is able to detect every single instance of every object class within an image. The challenge, then, is to select relevant objects. As the number of examples N increases, we would expect there to be fewer ways to select irrelevant objects and also satisfy every ranking constraint where those objects occur. While the “tree + people” example exists at the semantic level, there is also serious consideration for simple spurious patterns that the model could use to cheat. However, applying augmentation seems to relieve this risk.
>
> Adversarial Density Map: The purpose of our novel method is for the discriminator to structure the output of the generator to look like a density map, which another reviewer has highlighted as an important aspect of the paper. Our experiments have demonstrated empirically that our method improves results. We have a few hypotheses about why this should help. First, it explicitly connects the ranking task to the detection task, and forces each detection to correspond to a single countable unit. Second, by uniform sampling object locations across the density map, we encourage the output of the generator to seek spatially diverse density maps. While the objects are not uniformly spatially distributed in practice, they are still spatially distributed. Suppose there is some trivial pattern that occurs within an image ranking pair, such as a man in a yellow shirt occurring in one image and a man with a red shirt occurring in another image. The model could learn a trivial solution by pooling density over the colored shirts. However, a discriminator which selects for uniform distributions will prevent the model from pooling density over trivial features. Thus, the ranking loss and the uniform distribution loss together encourage the model to select against irrelevant objects.
>
> Annotation Cost: While it’s true that labeling images with their counts requires a smaller number of images to annotate in order to generate a target number of ranking pairs, there is a hazard involved, which is that using a smaller number of images implies a less diverse set of images used for training (even if the total number of target ranked image pairs is met).  For example, to collect 2000 ranked image pairs, we could either randomly sample 2000 image pairs from 500 images vs. explicitly counting the number of objects in 64 images to generate ~2000 (64*63/2=2016) image pairs. The former approach (our proposal) generates greater image diversity and reduces the risk of overfitting.
>
> Experiments: This is a good point. We agree that evaluation using these datasets would strengthen the paper. However, dense crowd counting (images with greater than 200 objects) is not the target for this method. But, we can run the experiments and include them in the experimental section to help quantify where the method works and where it fails. While this method does not necessarily generalize to dense crowd counting, we do have the option of dividing images to reduce the total number of objects per image.

---

### Official Review · Reviewer_MGT7 · 2021-11-03

**Correctness:** 3
**Technical Novelty And Significance:** 2
**Empirical Novelty And Significance:** 2
**Recommendation:** 3
**Confidence:** 4

**Details Of Ethics Concerns:**

If further annotation cost evaluation is needed, ethic reviews should be done.

**Main Review:**

Strengths:

The idea of using count-based ranking to train a weakly supervised object counting and localization method is interesting. The authors attempted to analyze the model performance vs the annotation burden. The overall writing is clear.


Weaknesses:

The comparison of the annotation burden is not convincing. Experiments were conducted under different settings. For example, annotation costs are referred from different works under different settings. The ranking cost should be relevant to the difference between two images. While, the experiment was conducted using randomly sampled image pairs. We don't know the count differences. Overall, the cost comparison needs more evidence.

The technical contribution is also limited.


------------post rebuttal--------------------
After reading the authors's rebuttal, I decide to keep my original score as 3. This work is interesting, but needs to be improved.

First, reducing annotation cost and improving model accuracy are the key motivations for weakly supervised localization. It is important to have a fair comparison between the proposed pair-wise ranking cost and other existing annotation costs to justify the reason of using the pair-wise ranking. The current version does provide some comparison, but it is not convincing. Different experimental settings may lead to significantly different cost estimations. I would suggest the authors do some annotation cost study in a fair setting.

Second, the performance comparison with existing methods is not convincing. As mentioned earlier, two factors (annotation cost and model accuracy) are used to evaluate weakly supervised methods. But, the annotation costs are all estimated using different sources. Besides, not only the ranking cost, but also the model optimization might be different given different pairs of ranked images. This paper only provides one set of randomly sampled image pairs, but does not analyzes how the model performance will be affected by the randomness. The authors mentioned that "Weber-Fechner law, which states that a human's ability to perceive a difference between the magnitude of two stimuli is related to the ratio between those stimuli". This is for the annotation cost, but empirical experiments are needed to verify if this is the case for model accuracy as well.

**Summary Of The Paper:**

This paper proposed an object counting and localization method which uses pairwise image ranking information for training. A count-based ranking objective is proposed for model optimization and an adversarial density map generation strategy is introduced to regularize the features of the network. The method was evaluated with multiple datasets.

**Summary Of The Review:**

The idea of using count ranking is interesting. The annotation cost comparison is not convincing and the work is lack of technical contribution.

---

> ### Author Response · Authors · 2021-11-20
> **Rebuttal**
>
> Hello Reviewer,
>
> Thank you for your review and your insights! We would like to address the concerns you have raised.
>
> With respect to the point about the technical contribution being limited, we would argue that this is a novel use of GANs on a novel version of the weakly supervised object counting problem. Further, we are the first to use inter-image ranking annotations to solve the weakly supervised counting problem. And we are attempting to tackle a fundamental underlying problem in the weakly supervised object counting space, which is that global labels underperform due to a lack of correspondence with local object features.
>
> To your point about the annotation burden being unconvincing, it’s true that we are pulling different annotation costs from different sources. There is, at present, no comprehensive source that analyzes the burden under various setups. To compensate for this, we provide a range where available and draw attention to the fact that we are working with estimates from different sources. The point of the paper isn’t to explicitly prove something about the true underlying nature of the burden of these various annotation types. We simply seek to provide motivation for this version of the weakly supervised object counting problem. In the limiting case, ranking two images requires a user to count the objects in both images. So, the ceiling for the annotation burden for ranking is the cost of counting annotations. Given that the cost for the counting problem is an order of magnitude [1,2,3] greater than the floor for the ranking problem, the annotation burden estimates should be sufficient to motivate the problem and give users a ballpark for what to expect. We don’t know of many examples of Weakly Supervised Counting and Weakly Supervised Object Localization papers that provide any information actually comparing the annotation burden between their respective fully and weakly supervised annotations. So, we have gone a step beyond many of these works by providing estimates, backed up by literature, for our method.
>
> With respect to the point about the relevance of the difference in the number of objects between two images, the more important measure is the ratio of the quantity of objects (not their absolute counts). This is known as the Weber-Fechner law, which states that a human's ability to perceive a difference between the magnitude of two stimuli is related to the ratio between those stimuli. We performed calculations for the Trancos and Penguins datasets. We will provide 3 metrics; the mean/std for the absolute difference for the object counts of the ranking pairs, the mean/std for the ratio of object counts between the smaller count and larger count, and the probability of selecting a pair that violates the Weber-Fechner ratio. Here, the Weber-Fechner ratio for object ranking is reported to be 10:11 or 0.91. For the Trancos dataset, we report a mean absolute difference of 15.3 objects with a std of 12.3 objects, and a mean ratio of 0.66 with a std of 0.20. Note that the ratio must be greater than 0.91 to violate the Weber-Fechner ratio. We also report a probability of 13.2% for selecting an image pair that violates the Weber-Fechner ratio. If we assume that an annotator presented with an image pair that violates this ratio will get 50% of these labels wrong, then we expect an error rate of 6.6% within the Trancos dataset. For the Penguins dataset, we report an mean absolute difference of 19.38 objects with a std of 20.0 objects, and a mean ratio of 0.52 with a std of 0.27. We also report a probability of 8.3% for selecting an image pair that violates the Weber-Fechner ratio and an expected error rate of 4.2%. To add further context, it is known that many popular image analysis datasets contain similar error rates. For example, it has been reported that the ImageNet test set has an error rate of 5.83% and the CIFAR-100 test-set has an error rate of 5.85% [4]. So the error rate would be comparable to other benchmarks, but with the context that errors for ranking pairs are made on examples that have similar object counts. So, 93.5 of the examples used during training contain image pairs that should be correctly labeled, regardless.
>
> [1] Hisham Cholakkal, Guolei Sun, Salman Khan, Fahad Shahbaz Khan, Ling Shao, and Luc Van Gool. Towards partial supervision for generic object counting in natural scenes. IEEE Transactions on Pattern Analysis and Machine Intelligence, 2020.
>
> [2] Pierre Pica, Cathy Lemer, Veronique Izard, and Stanislas Dehaene. Exact and approximate arithmetic in an amazonian indigene group. Science, 306(5695):499–503, 2004.
>
> [3] Justin Halberda and Lisa Feigenson. Developmental change in the acuity of the” number sense”: The approximate number system in 3-, 4-, 5-, and 6-year-olds and adults. Developmental psychology, 44(5):1457, 2008
>
> [4] Northcutt, Curtis G., Anish Athalye, and Jonas Mueller. "Pervasive label errors in test sets destabilize machine learning benchmarks." 2021

---

### Official Review · Reviewer_GhpV · 2021-11-08

**Correctness:** 3
**Technical Novelty And Significance:** 3
**Empirical Novelty And Significance:** 2
**Recommendation:** 6
**Confidence:** 4

**Main Review:**

The main contributions of the paper are important and interesting, the paper is well written and easy to follow, and the experiments seem to show the advantages of using the adversarial and rank loss. I still have few concerns that needs to be addressed:

1) Using GAN to learn structuring the output is interesting and important aspect of the paper. My question is if this approach has been used before in any previous work? Specifically, using GAN to structure the predictions like heatmap or segmentation mask where the "real" examples for training the GAN are synthetically generated.

2) I do not understand why only 2K pairs are sampled from each dateset? This is very small for all the datasets but specifically for Penguins with 82K images. If this is a saturation point for the accuracy it should be clearly mentioned in the paper otherwise the authors should provide the effect of increasing N in the final performance for some of these datasets.

3) Related the point (2), I also think it is important to show the effect of reducing the training set size for the state-of-the art methods and see how much their performance drops if fewer images are labeled. At the minimum, author can compare their method to the state-of-the-art when both method uses datasets with equal *annotation time*. Only then one can judge if using weak ranking labels is beneficial for learning to count.

4) What is the ratio of object counts in the sampled 2K pairs for each dataset (mean and std)? It is important to know this ratio is in the range in which human can easily do the ranking.

**Summary Of The Paper:**

The paper propose a learning to rank surrogate approach as surrogate for the problem of counting and localization in dense senses. It is claimed that this labeling approach significantly reduces the labeling time and it sufficient to learn a dense map which can be used for counting and localization. In addition to the pairwise ranking loss, the paper use an adversarial training strategy to make the predictions look more like dense maps.

**Summary Of The Review:**

Reviewer believes the paper has some interesting novelties but more ablation studies are necessary to support the points made in the paper. I am looking forward to reading the rebuttal and will adjust my score accordingly. I am willing to update my score to accept if the rebuttal addresses my concerns.


----------------- Post Rebuttal -------------------------

Thanks for addressing most of my concerns. I increased my score to 6 with regard to the rebuttal. I think the paper could be stronger with more experiments including dataset size vs. accuracy plots which show where the accuracy plots saturate.

Most of the points raised by the reviewers are just and important. However, I disagree with reviewer yR54 on the importance of using "Adversarial Density Map". As the reviewer mentioned, the problem of weakly supervised learning is typically ambiguous as the network might find to detect non-relevant features that still can decrease the final ranking loss. As this is also the case for other weakly supervised methods including object detection, I don't think the authors should be penalized for this. Instead, we need to focus on the novelties that can address some of these ambiguities. Adversarial density map loss put a meaningful constraint on the predictions to make them look like a mixture of gaussian among all the possible distributions that the network can generates. It is unlikely that it can solve all the ambiguities in the task but it should resolve some.

---

> ### Author Response · Authors · 2021-11-20
> **Rebuttal**
>
> Hello Reviewer,
>
> Thank you for all of the insight and feedback you have offered! We would like to address the 4 points that you have raised.
>
> To your first point, we have yet to find a paper that uses GANs to structure the output of a network to have the properties of density maps using a synthetic distribution. The most similar work to ours is the work in [1], where the authors use a discriminator to incorporate priors about the structure of the human body into the output of a pose estimation network. Our work uses synthetic density maps to encourage the network output to look like a density map. Their work uses real pose maps to learn a prior about the relationship between body parts. So far as we can tell, our work is a novel application of GANs.
>
> To your second and third points, we agree that we should include results for error rate and dataset size. The motivation behind choosing only 2K pairs was simply to minimize the experiment burden while demonstrating the effectiveness of the method. We had performed some experiments with this method previously and not found 2K pairs to be a saturation point, but we agree that the results would be stronger with this information. However, this method should not arbitrarily saturate at 2K pairs; imagine that we have a model which is already able to detect every instance of every object class within an image. Now, we merely need it to determine which of the detected objects are relevant. And suppose we have N ranking pairs which inform the model about which objects are relevant. If N is small then there are potentially many object selections that can satisfy the N ranking constraints. However, as N increases, the likelihood of violating a ranking constraint increases, which should select for models which ignore irrelevant objects.
>
> To your fourth point, we performed calculations for the Trancos and Penguins datasets. We will provide 3 metrics; the mean/std for the absolute difference for the object counts of the ranking pairs, the mean/std for the ratio of object counts between the smaller count and larger count, and the probability of selecting a pair that violates the Weber-Fechner ratio. Here, the Weber-Fechner ratio is the ratio between magnitudes of a pair of stimuli necessary to recognize a difference, and for object ranking it is reported to be 10:11 or 0.91. For the Trancos dataset, we report a mean absolute difference of 15.3 objects with a std.dev of 12.3 objects, and a mean ratio of 0.66 with a std.dev of 0.20. Note that the ratio must be greater than 0.91 to violate the Weber-Fechner ratio. We also report a probability of 13.2% for selecting an image pair that violates the Weber-Fechner ratio. If we assume that an annotator presented with an image pair that violates this ratio will get 50% of these labels wrong, then we expect an error rate of 6.6% within the Trancos dataset. For the Penguins dataset, we report an mean absolute difference of 19.38 objects with a std.dev of 20.0 objects, and a mean ratio of 0.52 with a std.dev of 0.27. We also report a probability of 8.3% for selecting an image pair that violates the Weber-Fechner ratio and an expected error rate of 4.2%. To add further context, it is known that many popular image analysis datasets contain similar error rates. For example, it has been reported that the ImageNet test set has an error rate of 5.83% and the CIFAR-100 test-set has an error rate of 5.85% [2]. So the error rate would be comparable to other benchmarks, with the additional context that errors for ranking pairs are made on examples that have very similar object counts. So, randomly flipping these examples wouldn’t be expected to have a large impact on the success of the method.
>
>
> [1] Chen, Yu, et al. "Adversarial posenet: A structure-aware convolutional network for human pose estimation." Proceedings of the IEEE International Conference on Computer Vision. 2017.
>
> [2] Northcutt, Curtis G., Anish Athalye, and Jonas Mueller. "Pervasive label errors in test sets destabilize machine learning benchmarks." (2021).

---

### Decision · Program_Chairs · 2022-01-20

**Decision:**

Reject

**Comment:**

Even though reviewers found some responses by the authors satisfactory, several concerns regarding the paper still remain. The authors are strongly encouraged to:

1) Explore how dataset size impacts accuracy.
2) Reason about annotation costs via empirical experiments.
3) Including benchmark datasets in experimental evaluations.